# Sub-Lobar Resection: The New Standard of Care for Early-Stage Lung Cancer

**DOI:** 10.3390/cancers15112914

**Published:** 2023-05-25

**Authors:** Benjamin E. Lee, Nasser Altorki

**Affiliations:** Division of Thoracic Surgery, Department of Cardiothoracic Surgery, Suite M404, Weill Cornell Medicine of Cornell University, 525 East 68th Street, New York, NY 10065, USA

**Keywords:** NSCLC, sub-lobar resection, segmentectomy, wedge resection, lobectomy

## Abstract

**Simple Summary:**

Lobectomy was shown to be superior to limited resection by the Lung Cancer Study Group in the surgical treatment of early staged non-small cell lung cancer (NSCLC) in 1995. Two recent non-inferiority randomized studies from the Cancer and Leukemia Group B co-operative group and the Japanese Clinical Oncology Group have compared sub-lobar resection with lobectomy and have established that sub-lobar resection was non-inferior to lobectomy for patients with clinical T1aN0 NSLC two centimeters or less. Here we review the design and results of these seminal trials and reflect on how their results may impact the treatment of NSCLC.

**Abstract:**

The Lung Cancer Study Group previously established lobectomy as the standard of care for treatment of clinical T1N0 NSCLC. Advances in imaging technology and refinements in staging have prompted a re-investigation to determine the non-inferiority of sub-lobar resections to lobectomies. Two recent randomized studies, JCOG 0802 and CALGB 140503, are reviewed here in the context of LCSG 0821. The studies confirm non-inferiority for sub-lobar resection (wedge or segmentectomy) compared to lobectomy for peripheral T1N0 NSCLC less than or equal to 2 cm. Sub-lobar resection should therefore be considered the new standard of care in this sub-set of patients with NSCLC.

## 1. Introduction

In 1995, the Lung Cancer Study Group reported the results of a randomized trial showing that a lobectomy was superior to a limited resection for patients with clinical T1N0 non-small cell lung cancer (NSCLC) [1]. In the decades since, lobar resections became the treatment of choice for patients with early-stage disease who have adequate cardiopulmonary function, while limited resections are generally restricted to patients who have compromised cardiopulmonary function [2,3,4].

Prompted by significant advances in imaging technology and refinements in TNM staging, both the Cancer and Leukemia Group B co-operative group (now part of the Alliance for Clinical Trials in Oncology) and the Japanese Clinical Oncology Group (JCOG) initiated non-inferiority randomized trials comparing lobar (LR) to sub-lobar (SLR) treatments in patients with clinical T1aN0 two centimeters or less in size [5,6]. The recently reported results from both trials established that sub-lobar (SLR) treatment was non-inferior to a lobectomy for patients with clinical T1aN0 NSCLC two centimeters or less in size. Here we review the design and results of these three seminal trials with particular emphasis on the similarities and differences between the two recent large trials and how their results may impact the treatment of early-stage lung cancer.

### 1.1. Lobectomy, Segmentectomy, and Wedge Resection Defined

The normal pulmonary anatomy accounts for 5 anatomic lobes (right upper, middle and lower, left upper and lower). Anatomic resection by lobectomy requires individual division of the lobar bronchus and the pulmonary arterial branches and pulmonary venous drainage to the corresponding lobe. By comparison, there are a total of 20 segments (10 on each side) and anatomic segmentectomy will divide the corresponding segmental bronchus, artery and vein. The parenchymal division will follow based on the segmental blood supply. Wedge resection is a non-anatomical resection that aims to remove the tumor and a surrounded margin of normal lung [7].

### 1.2. Pre-LCSG

The evolution of thoracic surgical resections has always been a trend from larger to smaller. Beginning in 1933 when Dr. Evarts Graham performed the first successful pulmonary resection for lung cancer by way of a pneumonectomy, it was but thirty years later, in 1962, that lobectomy was shown to offer similar results to pneumonectomy and became the new standard [8,9]. Fifteen years later, experiences with segmental lung resections were being published (of note, the first segmentectomy was performed by Drs. Churchill and Belsey in 1939 [10]) and segmentectomy was soon being compared to lobectomy [11]. A prominent non-randomized study examining 173 patients, published in 1994 comparing lobectomy (*n* = 105) and segmentectomy (*n* = 68) from Drs. Warren and Faber showed that for tumors greater than 3 cm, survival favored lobectomy [12]. However, in tumors less than 3 cm, there was no survival advantage. Locoregional recurrence was 22.7% after segmentectomy compared to 4.9% after lobectomy. Interestingly, a similar study was already underway with results published the very next year. This study, in contrast, was a randomized, prospective study put forth by the Lung Cancer Study Group (LCSG).

### 1.3. LCSG 0821

The LCSG 0821 was a prospective, multi-institutional randomized trial designed to show the equivalence of SLR resections to lobectomies [1]. Patients were registered if they had a suspected tumor measuring 3 cm or less in all dimensions on posteroanterior and lateral chest roentgenograms. Patients were randomized to lobectomy or limited resection (wedge or segment) after intraoperative confirmation of node negative disease in the segmental, hilar and mediastinal lymph nodes. Enrollment began in February 1982 and was completed in November 1988. Sixty four percent of screened patients (495/771) were not randomized, most commonly due to a finding of benign disease (24.5%, 189/771). In addition, 8.7% (67/771) were not randomized due to N1/N2 intra-operative upstaging. After excluding 29 patients due to major protocol violations, 247 randomized patients were analyzed for the trial endpoints; 125 who underwent lobar resection and 122 patients who underwent a limited resection (40 wedge resections and 82 segmentectomies). Table 1 illustrates the patient demographics and clinical characteristics. Patients in the LCSG were predominantly men (61%) with a smoking history (95% former or current). The investigators reported a 30% increase in the overall death rate (*p* = 0.088) and a 50% increase in observed death with cancer rate (*p* = 0.094) in the limited resection group. Given the trial design, the investigators declared both endpoints statistically significant in favor of lobectomy. The overall recurrence rate was 75% higher in the limited resection group (*p* = 0.02). This was mainly attributed to a 3-fold higher locoregional recurrence in patients who had a limited resection (*p* = 0.008) since the distant recurrence rates were similar between the two groups. In addition, locoregional recurrence after wedge resections were twice that reported after segmentectomy. Pulmonary function differences in expiratory flow rates favored limited resection at six months but had decreasing significance at 12 and 18 months. Although the lung cancer study group trial determined the standard of surgical care for decades, the trial had a critical methodological flaw. The trial was designed as an equivalency trial, a design rarely recommended for therapeutic trials, with a 90% power to detect a 1.8 -fold difference in median survival or a 1.85-fold difference in median time to recurrence, both treatment effects in favor of lobectomy. This wide equivalence margin was selected to accommodate a smaller sample size and shorter follow-up. The trial was thus underpowered to demonstrate a clinically reasonable equivalency or non-inferiority between lobar and limited resections. 

### 1.4. Lobectomy vs. Segmentectomy

In the years following the results reported by the LCSG, segmentectomy was typically reserved for patients with limited pulmonary reserve or advanced age. However, with technological advancements in imaging modalities (such as high-resolution CT, and PET/CT), retrospective investigations comparing segmentectomy with lobectomy were ever present. In 2013, Yendamuri et al. reported a decreased benefit of lobectomy over sublobar resection based on a temporal trends outcome from a SEER (surveillance epidemiology end results) database study [13]. In this study, 8797 patients were grouped into 3 time periods (1988–1998, 1999–2004, 2005–2008) and assessed. Interestingly, the proportion of female patients, patients with tumors less than 2 cm, and proportion of lower grade tumors all steadily increased over each time period. In the latest time period, there were no significant difference in overall or disease-free survival rates between sublobar resections and lobectomies. The authors postulated possible theories including biological changes over time in NSCLC and better outcomes favoring women with NSCLC. In a separate study by Razi et al., also generated from the SEER database in 2016, 1640 patients over the age of 75 who underwent sublobar resections or lobectomies were examined [14]. Altogether, there were 1051 lobectomies, 119 segmentectomies, and 470 wedge resections. Lobectomy was found to confer a survival advantage in T1b (2–3 cm) tumors but in patients with T1a tumors (<2 cm), there was no significant difference in risk adjusted 5-year cancer specific survival rates for patients undergoing wedge resection, segmentectomy (hazard ration 1.009, *p* = 0.972), or lobectomy (hazard ration 0.98, *p* = 0.908). The authors concluded that sublobar resection was non inferior to lobectomy for T1aN0M0 NSCLC. Despite these studies reported from the SEER database both favoring sublobar resection, a third SEER database study by Dai et al. published in 2016 examined 15,760 patients with 2 cm or smaller T1aN0M0 and segregated the patients into groups based on tumor size (<1 cm, 1–2 cm) [15]. Lobectomy showed improved survival compared to sublobar resection in both groups and multivariate analysis showed that wedge resection was an independent risk factor of survival. The authors concluded that lobectomy should remain the standard and that sublobar resection should be reserved only for patients who could not tolerate lobectomy. A separate retrospective study examining the NCDB (National Cancer Database) in 2015 by Khullar et al. examined 28,241 patients (19,718 lobectomy, 7297 wedge resection, 1226 segmentectomy) and demonstrated significant worsened overall survival for T1a NSCLC patients undergoing wedge and segmental resections compared to lobectomy [16]. Patients undergoing sublobar resections were older with slightly smaller tumors. Furthermore, sublobar resections were associated with significantly fewer lymph node stations examined and a higher rate of a positive margin. The authors concluded that decreased upstaging due to poor lymph node harvesting was the likely cause of worsened survival in the sublobar group. In 2017, another study born out of the NCDB by Cox et al. examined 1991 patients with clinical stage 1 adenocarcinoma with lepidic histology [17]. In this study, 1544 patients underwent lobectomies and 447 underwent sub-lobar resections. Again, sub-lobar patients were older, female, had higher Charlson/Deyo comorbidity scores, and smaller tumors with lower T-status. 94.6% of patients undergoing a lobectomy had a pathologic lymph node evaluation compared to only 45% of patients undergoing sub-lobar resections. In the univariate analysis, lobectomy was associated with a significant survival advantage with a median survival of 9.2 years compared to 7.5 years in the sub-lobar group (*p* = 0.022). However, within the sub-lobar group, patients who had a lymph node assessment were found to have a significantly improved 5-year survival compared to those who did not (71.1% vs. 65.1%). Accordingly, when sub-lobar patients with a lymph node assessment were compared to lobectomy patients, no significant difference in survival was seen (HR 0.93, *p* = 0.56).

During this same time period, the Early Lung Cancer Action Project (ELCAP) had an ongoing prospective database containing data on patients being screened for lung cancer from 1993 to 2011 [18]. NSCLC was identified in 347 patients who underwent surgical resection (294 lobectomy, 16 segmentectomy, 37 wedge resection). The groups were well balanced in their demographics and clinical variables including age, gender, smoking history, histology, tumor size, and pathologic T and N stages. The only observed difference was a significantly fewer number of lymph nodes harvested in the sub-lobar group. Evaluation of 10-year overall survival rates for the 53 sub-lobar patients was 85% compared with 86% in the 294 lobectomy patients (*p* = 0.86). For tumors less than 20 mm, the 10-year survival rates were 88% and 84%, respectively (*p* = 0.45). All of these studies added to the uncertainty of the utility of sub-lobar resection in early-stage NSCLC and referenced the ongoing JCOG 0802 and CALGB 140503 studies with hope for a final resolution.

### 1.5. JCOG 0802

JCOG 0802 was a randomized, multi-institutional, non-inferiority trial designed to test the hypothesis that in patients with clinical stage IA small-sized (≤2 cm) NSCLC, segmentectomy is non-inferior to lobectomy for the primary endpoint of overall survival [5]. The secondary endpoints included postoperative respiratory function and relapse-free survival. Patients were enrolled between August 2009 and October 2014. Eligible patients had tumors ≤ 2 cm located in the outer third of the lung on computerized tomography (CT). One thousand three hundred and nineteen patients were identified and two hundred and thirteen (16.1%) failed intraoperative randomization mostly due to the finding of benign disease (6.3%, 83/1319) or N1/N2 upstaging (0.2%, 3/1319). In total, 1106 patients were randomized to either lobectomy (*n* = 554) or segmentectomy (*n* = 552). Patients were predominantly male (53%) and 44% were never-smokers. In addition, 97.9% (1083/1106) had an ECOG (Eastern Cooperative Oncology Group) performance score of zero. More than 90% of patients had adenocarcinoma. The study found a five-year overall survival rate of 94.3% after segmentectomy compared to 91.1% with lobectomy. Segmentectomy was found to be both non-inferior (*p* < 0.0001) and superior (*p* = 0.0082) to lobectomy. Total lung cancer deaths were 26/552 in the segmentectomy group and 28/554 in the lobectomy group. However, other deaths were markedly higher in the lobectomy group (52 vs. 27) and were mostly due to other cancers including 2nd primary lung cancer (31 vs. 12). Five-year disease-free survival was 88% with segmentectomy and 87.9% with lobectomy although local relapse was significantly higher with segmentectomy (10.5% vs. 5.4%, *p* = 0.0018). Patients with relapse were more likely to be alive at five years in the segmentectomy group (68%) compared to the lobectomy group (49%), more likely to receive treatment for relapse (93% vs. 80%, respectively), and re-operation for 2nd primary lung cancer (89% vs. 63%, respectively). At six months, FEV1 was reduced 10.4% vs. 13.1% favoring segmentectomy (*p* < 0.0001).

### 1.6. CALGB (Alliance) 140503

CALGB/Alliance 140503 was also a prospective, multicenter randomized non-inferiority phase three trial which randomized patients to either lobectomy or SLR [6]. Randomization was stratified by tumor size (<1.0 cm, 1.0–1.5 cm, >1.5–2.0 cm), smoking history (never, former, current) and histology (squamous carcinoma, adenocarcinoma, other). The primary endpoint was disease-free survival. Secondary endpoints included overall survival, locoregional recurrence, and pulmonary function. Eligible patients had peripheral lung nodules with a solid component 2 cm or less in size on preoperative computerized tomography (CT) presumed or confirmed to be NSCLC; located in the outer third of the lung. Patients with pure ground-glass opacities or pathologically confirmed N1 or N2 disease were not eligible. Eligible patients were registered and intraoperatively randomized after confirmation of NSCLC diagnosis (if not already done) and pathological N0 status by frozen section examination of at least two mediastinal nodal stations and a major hilar station. The modality of SLR (wedge resection vs segmentectomy) was left to the discretion of the surgeon. The trial was activated in June 2007 and closed to accrual on March 13, 2017. One thousand and eighty patients were registered to the trial, of whom 35% (383/1080) were not intraoperatively randomized. Of these, 16.3% of patients failed randomization due to not having NSCLC and 6.4% had N1/N2 disease. A total of 697 patients were successfully randomized to either SLR (*n* = 340) or lobectomy (*n* = 357). Within the SLR group, 201 patients (59.1%) underwent a wedge resection and 129 (37.9%) underwent segmentectomy. Patients were predominantly women (57%), had a positive history of smoking (91% former or current), and 73.6% (513/697) were ECOG 0. Sixty three percent of patients had adenocarcinoma, 14.1% had squamous cell carcinoma, and 22.2% had “other” histology.

For the primary endpoint of disease-free survival, SLR was non-inferior to lobectomy (HR 1.01, 90%CI: 0.83, 1.24). The five-year disease-free survival rate was 63.6% in the SLR arm and 64.1% in the lobectomy arm (*p* = 0.02 for non-inferiority). The five-year overall survival rate was 80.3% vs. 78.9%, respectively. Overall disease recurrence was 30.4% (102/336) in the sublobar group compared with 29.3% (103/351) in the lobar group and locoregional recurrence was higher in the sub-lobar group (13.4% vs. 10.0%). The five-year recurrence free survival rate was 70.2% after SLR compared with 71.2% after lobectomy and lung cancer related deaths were higher in the lobectomy group (55 vs. 46). FEV1 reduction at six months was 6% vs. 4% favoring segmentectomy.

### 1.7. JCOG vs. CALGB

Although JCOG 0802 and CALGB 140503 have both met their primary endpoints proving that SLR is non-inferior to lobectomy for overall and disease-free survival respectively, there are important differences between the two trials in design, demographics, and patients’ outcomes (Table 1). First, patients were selected for JCOG 0802 if the total tumor size on a CT scan was 2 cm, and therefore by design the trial included patients with part solid nodules that constituted nearly 50% of the trial population. In contrast, in CALGB 140503 patient selection was based on the size of the solid component and therefore the trial likely included very few patients with any ground glass component. Second, the two trials also differed in the type of SLR employed, since only anatomical segmentectomy was allowed in the Japanese trial while both segmentectomy and wedge resection were permitted in the North American trial. Finally, the study population in JCOG 0802 is markedly different compared to that in CALGB 140503. In JCOG 0802, nearly half of the patients were never smokers, almost the entire cohort had an ECOG score of zero, and over 90% of patients had adenocarcinoma. In contrast, in CALGB 140503, the majority of patients were either current or former smokers (91%), 26% had an ECOG performance score of 1/2 and 36% had either squamous cell cancer (14%) or “other” cell types (22%); a category that included large cell, pleomorphic, adeno-squamous and neuro-endocrine carcinoma. In aggregate these differences may have contributed to the excellent overall survival reported by JCOG 0802 investigators, which were 91% and 94% after LR and SLR, respectively, and the remarkable relapse free survival rate of 88% in both arms of the trial. In contrast, disease free survival, the primary endpoint of the CALGB trial, was 64% after LR and 63% after SLR. Overall survival was 79% after LR and 80% after SLR (Table 2). Finally, despite a higher local recurrence rate in the segmentectomy group, five-year survival for patients undergoing either segmentectomy or lobectomy in JCOG 0802 both exceeded 90%, which is well above the current world-wide survival for stage 1A NSCLC [19]. The results from JCOG 0802 are truly fantastic but may be limited to a specific demographic. It is also possible that the differences in overall and disease-free survival between the two trials are due to ethnic base differences in disease biology between a predominantly white population of mainly Caucasian descent and patients of Asian ancestry. Multiple studies have demonstrated ethnic differences in survival outcomes in NSCLC with survival advantages seen in patients of Asian descent [20,21,22,23].

### 1.8. Wedge vs. Segmentectomy

There have been several retrospective studies that have attempted to answer this ongoing debate in the past decade alone. In 2013, Smith and Swanson et al. examined the SEER database to identify patients with stage 1A NSCLC < 3 cm who underwent wedge resection (*n* = 1568) or segmentectomy (*n* = 378) between 1998 and 2006 [24]. Propensity score matching was performed and included adjustments for the number of lymph nodes evaluated. Segmentectomy was found to have a significantly improved overall survival rates (HR: 0.80) and lung cancer specific survival rates (HR 0.72) over wedge resection. Subsequently in 2016, Altorki et al. performed a retrospective review examining patients undergoing wedge resection (*n* = 160), or segmentectomy (*n* = 129) for cT1N0 NSCLC [25]. Segmentectomy was associated with a higher likelihood for lymph node sampling/dissection (95% vs. 70%), more lymph node stations sampled (3 vs. 2) and lymph nodes removed (7 vs. 4). There was, however, no significant difference in local recurrence or five-year disease-free survival rates. In 2019, a study from Japan by Tsutani et al. examined the outcomes of 99 patients with NSCLC who underwent surgical resection (Wedge *n =* 60, segmentectomy *n* = 39) [26]. Severe complications were more common in the segmentectomy arm, but overall survival rates and recurrence-free survival rates were not significantly different. A second study from Japan by Mimae et al. in 2021 looked specifically at wedge resection or segmentectomy in octogenarians with solid predominant NSCLC < 2 cm [27]. Three-year overall survival was not significantly different and may even favor wedge resection. Another study in 2021 from Chiang et al. examined patients from Taiwan who underwent sublobar resection (wedge resection *n* = 810, segmentectomy *n* = 192) for cT1N0 lung adenocarcinoma between 2011 and 2017 [28]. After propensity matching, there was no significant difference in overall survival or DFS. However, in patients with tumors greater than 2 cm, segmentectomy was found to have an improved DFS (*p* = 0.039). Finally, a recent study by Akamine et al. again retrospectively examined 720 patients undergoing sub-lobar resection (segmentectomy *n* = 479, wedge resection *n* = 241) for clinical stage 1 NSCLC from 2017 to 2020 [29]. An adequate surgical margin was more associated with segmentectomy (71.4% vs. 59.5%, *p* = 0.002) and segmentectomy showed a significantly improved recurrence-free survival rate (hazard ratio 2.7, *p* < 0.001).

This ongoing debate of wedge resection versus segmentectomy is not definitively answered by JCOG 0802 nor CALGB 140503, as neither were intended to address the question. However, it is important to note that wedge resection for NSCLC remains the most common type of lung cancer resection performed in North America and Europe today [30,31]. Overall, the results of CALGB 140503, where 58% of all SLR utilized wedge resection, appear to challenge the prevailing view that wedge resections are inherently inadequate oncological procedures. Although, an analysis of the results after wedge resection versus segmentectomy are eagerly awaited, it is highly unlikely that there will be a meaningful difference between the two modalities of SLR. In the meantime, it is important to emphasize that these results are only applicable to a highly selected group of patients who present with peripheral tumors two centimeters or less in size and in whom the absence of nodal disease had been confirmed by pathological examination of lymph nodes from at least one major hilar and two mediastinal nodal stations. Additionally, the trial protocol of CALGB140503 emphasized the need for obtaining an “adequate” resection margin when preforming a wedge resection and strongly encouraged intraoperative pathological assessment of the resection margin. To that end, an “adequate” margin was arbitrarily defined by the investigators as a two-centimeter margin around the tumor or a margin equivalent to the clinical tumor size.

Interestingly the authors of JCOG 0802 separately discuss the differences between a “simple” and “complex” segmentectomy and reported a significant increase in pulmonary complications associated with complex segmentectomy [32]. Practically speaking, the location of the tumor typically dictates the surgical options, especially when evaluating a patient for SLR. Some lesions are in ideal locations for a single anatomic (simple) segmentectomy, while others may straddle an intersegmental plane and pose technical challenges for simple segmentectomy, thereby requiring a complex segmentectomy [33,34]. Based on results from CALGB 140503, treatment by wedge resection may offer an alternative option with perhaps a similar benefit of decreased morbidity and mortality when compared with complex segmentectomy. All told, the results from both JCOG 0802 and CALGB 140503 offer significant evidence that sublobar resection, either by segmentectomy or wedge resection, is an acceptable treatment option for otherwise fit patients with peripheral stage IA NSCLC < 2 cm. Current randomized studies include the AWESOME trial/JCOG2109 which compares segmentectomy and wedge resection in predominantly solid < 2 cm stage 1A NSCLC in elderly (>80 years) patients who are not candidates for lobectomy. In addition, another similar study (clinical stage 1A NSCLC) in high-risk operable patients (ANSWER trial/JCOG1909) is also forthcoming.

### 1.9. Future Directions

Despite the questions that have been answered by both JCOG 0802 and CALGB 140503, results from these studies invariably raise new questions. Especially with CALGB 140503, if the five-year disease-free survival after either sublobar resection or lobectomy is only around 64% and locoregional recurrence exceeds 10%, should we be looking at examining the role of neo-adjuvant or adjuvant therapy in these patients? Recent data from Checkmate 816 and IMpower010 have demonstrated significantly improved survival rates in the combined use of chemotherapy and immunotherapy in either the neoadjuvant or adjuvant setting, respectively, in patients with stage 1B to 3A NSCLC [35,36]. Despite a dearth in evidence to support the use of chemotherapy alone in this setting, it may only be a matter of time before combination immunotherapy with or without chemotherapy before or after surgery is the new paradigm for treatment in stage 1A NSCLC.

In addition, it is inevitable that some may advocate for non-surgical ablative therapies including stereotactic body radiotherapy (SBRT) [37,38]. However, it is important to emphasize that patients successfully randomized to sublobar resections had both intraoperative lymph node sampling to decrease the risk for occult nodal disease as well as verification of a negative surgical margin of 2 cm or a distance equal to the size of the tumor. As it stands, there still is no prospective trials that have shown non-inferiority, equivalence, or superiority of SBRT to surgical resection for stage 1A NSCLC.

## 2. Conclusions

JCOG 0802 and CALGB 140503 are congruent studies offering solid evidence of the non-inferiority of SLR to a lobectomy for peripheral stage 1A NSCLC. Confirmation of the absence of nodal metastases by intra-operative pathological or cytological examination is an essential component of these treatment strategies and should not be replaced by imaging modalities with variable sensitivity. Previous retrospective studies also maintain a common theme that sublobar resections with adequate lymph node harvesting produces survival rates similar to lobectomy in these patients. Therefore, ensuring an adequate negative margin and performing a thorough lymph node assessment remain an essential requirement for a successful sublobar resection. Looking forward, the new horizons facing the treatment of small lung tumors offer much promise and much uncertainty. With the development of advanced navigational bronchoscopy, catheter-based ablations will compete with stereotactic radiosurgery and surgical resection for treatment in this space. Ultimately, as this review has shown, accurate staging remains tantamount to any future studies comparing these techniques.

## Figures and Tables

**Table 1 cancers-15-02914-t001:** Trial Design, Patient Demographics and Clinical Characteristics.

	LCSG 0821	JCOG 0802	CALGB 140503
Trial Design	equivalence	non-inferiority	non-inferiority
Enrollment	771	1319	1080
Randomized	276	1106	697
Ineligible after randomization	29	8	
Limited Resection			
Wedge	40	1	201
Segmentectomy	82	544	129
Lobectomy	125	553	357
Adenocarcinoma	184 (74.5)	1003 (90.7)	444 (63.7)
Squamous	63 (25.5)	75 (6.8)	98 (14.1)
Other		18 (2.5)	155 (22.2)
Male	149 (60.6)	583 (52.7)	297 (42.6)
Female	126 (39.4)	523 (47.3)	400 (57.4)
Smoker	234 (95.1)	616 (55.7)	634 (91)
Never smoker	12 (4.9)	490 (44.3)	63 (9)
Performance			
Karnofsky 10/ECOG 0	141 (57.3)	1083 (97.9)	513 (73.6)
Karnofsky < 10/ECOG 1/2	105 (42.7)	23 (2.1)	184 (26.4)

**Table 2 cancers-15-02914-t002:** Overall Survival, Disease Free Survival, Locoregional Recurrence.

	LCSG (Rate per Person/Year)		JCOG (5-Year)		CALGB (5-Year)	
	SLR	Lobectomy	Segmentectomy	Lobectomy	SLR	Lobectomy
Overall Survival	0.117	0.089	94.3	91.1	80.3	78.9
Disease Free Survival	0.073	0.049	87.9	88	63.6	64.1
Locoregional Recurrence	0.06	0.02	11	5	13.4	10

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
