# Peer review of "Sub-Lobar Resection: The New Standard of Care for Early-Stage Lung Cancer"

_cancers, 2023, doi:10.3390/cancers15112914_

Round 1

Reviewer 1 Report

Dear authors 

the article is well written and open unresolved questions which need an answer. I agree with the authors. As the authors cited the first lung successfully pneumonectomy in 1933 by Dr. Evarts Graham, my personal comment is that the authors should also give merit to those who performed the first anatomical segmentectomy in 1939 (although it was not for lung cancer but for bronchiectasies). 

Churchill  ED, Belsey  R. Segmental pneumonectomy in bronchiectasis: the lingula segment of the left upper lobe. Ann Surg. 1939;109(4):481–499.   Thank you for sending this excellent paper to Cancers

Author Response

Thank you for your kind remarks and for your added reference for the first anatomical segmentectomy.   We have included this into our paper.

Reviewer 2 Report

The authors presented an exhaustive systematic review of published data evaluating, retrospectively or prospectively, lobectomy  versus sublobar resection for stage IA NSCLC. The review is well-written, and the results are fairly well-discussed.

On a methodological point-of-view, the authors should comply with the PRISMA (http://prisma-statement.org/). In the present state, the reader is not informed about many quality criteria needed when reporting systematic review (such as search equation, databases, selection of studies). Risk of bias in retrospective studies should be highlighted by the discussion.

A meta-analysis of the three main prospective trials (LCSG, CALGB, JCOG) may be very interesting even the first study suffered from the poor disease stage work-up and surgical techniques used forty years ago.

The authors discussed the differences in patients' demography and disease characteristics that made the comparison between these two non-inferiority studies difficult.

In my opinion, the analysis of the JCOG study is not complete.

The study involved more than 500 patients operated on per group. This represents the work of 70 Japanese centres. The accrual was completed in 2014 and therefore there is a median monitoring of more than 7 years. All this gives an impression of the robustness of the study.

However, reading the methodology reveals a large number of exclusion criteria. For instance, the segmentectomy should not concern more than two segments.

Patients with interstitial lung disease, severe emphysema, or diabetes were non-eligible. They were also excluded if there were several synchronous and even metachronous tumors. This last point is a real limitation insofar as second cancers are frequent.

All this reduces the population concerned. In addition, the study imposed a methodology that is probably not possible everywhere: in order not to make an inclusion wrongly, the second stage of the inclusion was done per-operatively.

The protocol required extemporaneous histopathologic control of the malignancy (the diagnosis of which is therefore not known preoperatively), then once the patient has been randomized to the segmentectomy arm, if the surgical margin was less than the maximum diameter of the tumor or less than 20 mm, extemporaneous diagnosis on frozen specimens or cytological examination was mandatory to confirm the absence of tumor on the slice before closing the chest wall.

If the margin was positive, an additional partial resection was mandatory.

Overall survival is therefore better in the segmentectomy group. However, there is no benefit on the preservation of respiratory function. In point of fact, the hypothesis was that segmentectomy saves at least 10% more FEV1 than lobectomy. This is not the case because the difference between the two groups is only 3.5% one year after the resection. In addition, the locoregional recurrence rate was twice as high in the segmentectomy group (11%) than in the lobectomy group (5%, p <0.002).

So why this apparent contradiction: more local recurrences, but better overall survival for segmentectomies? Probably, because mortality related to cardiovascular accidents, respiratory failure or new neoplasms was significantly higher in the lobectomy group.

Author Response

The authors presented an exhaustive systematic review of published data evaluating, retrospectively or prospectively, lobectomy  versus sublobar resection for stage IA NSCLC. The review is well-written, and the results are fairly well-discussed.

    -- Thank you for your kind remarks.

On a methodological point-of-view, the authors should comply with the PRISMA (http://prisma-statement.org/). In the present state, the reader is not informed about many quality criteria needed when reporting systematic review (such as search equation, databases, selection of studies). Risk of bias in retrospective studies should be highlighted by the discussion.

   -- Our paper was not intended to be a systematic review nor a meta-analysis.   The intention was to discuss the only 3 published randomized studies which have compared lobectomy and sub-lobar resection for early-stage lung cancer.  Although other retrospective studies are mentioned, they are only used as background leading up to the randomized studies.

A meta-analysis of the three main prospective trials (LCSG, CALGB, JCOG) may be very interesting even the first study suffered from the poor disease stage work-up and surgical techniques used forty years ago.

   -- Indeed, a meta-analysis might be interesting but as you have stated, there are major differences in time and patient population between the studies which would limit the effectiveness of such a study.

The authors discussed the differences in patients' demography and disease characteristics that made the comparison between these two non-inferiority studies difficult.  In my opinion, the analysis of the JCOG study is not complete.  The study involved more than 500 patients operated on per group. This represents the work of 70 Japanese centres. The accrual was completed in 2014 and therefore there is a median monitoring of more than 7 years. All this gives an impression of the robustness of the study.  However, reading the methodology reveals a large number of exclusion criteria. For instance, the segmentectomy should not concern more than two segments.  Patients with interstitial lung disease, severe emphysema, or diabetes were non-eligible. They were also excluded if there were several synchronous and even metachronous tumors. This last point is a real limitation insofar as second cancers are frequent.  All this reduces the population concerned. In addition, the study imposed a methodology that is probably not possible everywhere: in order not to make an inclusion wrongly, the second stage of the inclusion was done per-operatively.  The protocol required extemporaneous histopathologic control of the malignancy (the diagnosis of which is therefore not known preoperatively), then once the patient has been randomized to the segmentectomy arm, if the surgical margin was less than the maximum diameter of the tumor or less than 20 mm, extemporaneous diagnosis on frozen specimens or cytological examination was mandatory to confirm the absence of tumor on the slice before closing the chest wall.  If the margin was positive, an additional partial resection was mandatory.  Overall survival is therefore better in the segmentectomy group. However, there is no benefit on the preservation of respiratory function. In point of fact, the hypothesis was that segmentectomy saves at least 10% more FEV1 than lobectomy. This is not the case because the difference between the two groups is only 3.5% one year after the resection. In addition, the locoregional recurrence rate was twice as high in the segmentectomy group (11%) than in the lobectomy group (5%, p <0.002).  So why this apparent contradiction: more local recurrences, but better overall survival for segmentectomies? Probably, because mortality related to cardiovascular accidents, respiratory failure or new neoplasms was significantly higher in the lobectomy group.

  -- We thought to answer these comments as a single section given that the questions were at the end.   Again, thank you very much for your comments.  

  1. With regards to the JCOG study, your observations are valid and we have attempted to highlight many of these within the paper. However, to address some of your specific points:
    1. The exclusion criteria in the JCOG study are similar to exclusion criteria when compared even to the CALGB study as CALGB also did not allow for patients with second primary lung cancers or any other malignancy within 3 years of the study period.  The CALGB study also excluded patients with a history of chemotherapy and/or radiation therapy.  We also agree that the study restrictions imposed by JCOG do limit its 'real world' applicability and we have highlighted this in our paper.  We do feel that the relative imbalance in histology (predominant adenocarcinoma) and high proportion of part-solid tumors are what really limits the applicability of the JCOG data.   It is also understandable the restrictions on severe emphysema and ILD given the need to compare lobectomy and sub-lobar resection as patients with these conditions may not have sufficient lung function to tolerate lobectomy.  As far as histopathological control and the intra-operative assessments on margin, this methodology was very similar to that of CALGB, where tumors also required intra-operative diagnosis and verification of margin status.  Again, having intraoperative margin assessment may not be a practical 'real world' 
    2. option, but in the setting of a clinical trial this was an important aspect to ensure completeness of resection.
    3. Secondly, the differences in FEV1 seen in the JCOG study were similar to both the LCSG and CALGB studies, with improvements favoring segmentectomy but perhaps not clinically significant.
    4. Lastly, we did note in our review that local recurrence rates with segmentectomy were significantly higher than with lobectomy.  As you have already stated, the segmentectomy patients all had a higher re-intervention rate for local recurrence or second primary.  This was all mentioned in our review.   We also postulated that improved survival rates in the JCOG study may be cultural as well.

Thank you again for your comments.

Reviewer 3 Report

I congratulate the authors with this exceptional and timely work, which should encourage surgeons around the world to modify their practice.

Author Response

We thank you for your kind comments.

Reviewer 4 Report

Very interesting review that clarifies and analyzes the data of papers related to the comparison, in terms of survival and disease-free interval, of patients with early lung cancer undergoing major or minor lung resection. The list of bibliographic references also appears complete.

Author Response

Thank you for reviewing our paper and for your comments.